# The Role of Sensor Technologies in Estrus Detection in Beef Cattle: A Review of Current Applications

**DOI:** 10.3390/ani15152313

**Published:** 2025-08-07

**Authors:** Inga Merkelytė, Artūras Šiukščius, Rasa Nainienė

**Affiliations:** Department of Animal Breeding and Reproduction, Animal Science Institute, Lithuanian University of Health Sciences, R. Zebenkos 12, 82317 Baisogala, Lithuania; arturas.siukscius@lsmu.lt (A.Š.); rasa.nainiene@lsmu.lt (R.N.)

**Keywords:** beef cattle, reproduction, estrus detection, sensor-based monitoring, infrared thermography

## Abstract

In beef cattle farming, detecting the right time for insemination is crucial but often difficult when using traditional observation methods. This article explores how new technologies like sensors and thermal cameras help farmers identify when cows are ready to breed. These tools monitor changes in movement and body temperature, which are early indicators of estrous. With these systems, farmers can act quickly, improving the chances of successful insemination. This means better productivity, healthier herds, and more sustainable farming practices. Even though the tools can be expensive and require some training to use, they are becoming more accessible and may soon become common on cattle farms.

## 1. Introduction

Modern strategies for managing beef cattle reproduction face growing challenges due to the increasing global demand for beef. Population growth, rising economic levels, and changing dietary habits have significantly contributed to a nearly 58% rise in global meat consumption over the past two decades [1]. This trend underscores the necessity of improving reproductive management methods to enhance productivity and reduce operational costs. However, beef cattle production efficiency remains among the lowest compared to other livestock species, highlighting the need for innovative technological solutions to optimize reproductive performance [2].

The main goal in reproductive management is to ensure that each cow produces one calf per year, as this is critical for maintaining herd productivity and economic viability [3,4]. Achieving this goal requires precise control of the breeding period, typically occurring between 65 and 85 days postpartum. Failure to achieve fertilization within this window extends the calving interval, negatively impacting herd profitability. One of the key challenges to reproductive success is silent estrus, a condition in which ovulation occurs without observable behavioral signs. This phenomenon is particularly prevalent during the first estrous cycle after calving, affecting approximately 50–80% of cows and complicating the timely scheduling of artificial insemination [5].

Traditional reproductive management methods, which primarily rely on visual estrus detection and artificial insemination, become increasingly ineffective as herd sizes grow. Subjective diagnostic errors further exacerbate the inefficiencies associated with these conventional approaches [6]. To address these limitations, advanced monitoring technologies are being employed, playing an increasingly vital role in modern cattle reproduction management. Sensor-based systems, such as accelerometers, allow for the real-time monitoring of cattle activity levels, providing objective data used to detect estrus and reducing reliance on visual observation. Additionally, infrared thermography (IRT) offers a non-invasive method for identifying estrus onset by detecting subtle fluctuations in body temperature, thereby improving the accuracy of artificial insemination timing and overall reproductive efficiency [7].

Despite the evident benefits of these advanced technologies, their adoption is hindered by financial and technical barriers. High initial investment costs, coupled with the need for specialized knowledge regarding data interpretation, may limit widespread implementation, particularly among small and medium-sized farms. However, continued advancements in technology and the development of user-friendly, cost-effective solutions have the potential to enhance accessibility and scalability, thereby improving reproductive efficiency in the beef cattle industry.

This review examines the latest advancements in smart technologies for beef cattle reproduction, with a particular focus on sensor-based monitoring systems and infrared thermography. It explores commercially available technologies and discusses the scientific literature assessing the prospects of emerging systems. Additionally, this review highlights the challenges associated with these technologies and evaluates their potential to significantly enhance reproductive efficiency and sustainability in beef cattle production.

## 2. Literature Review Methodology

A systematic literature search was conducted in March 2025 using the following electronic databases: Google Scholar and Scopus. The search was limited to peer-reviewed scientific articles published in English. The literature search was conducted using the following keywords: (beef cattle reproduction OR beef cattle estrus detection) AND (beef cattle OR cow reproduction) AND (sensor OR accelerometer OR pedometer OR heat detection OR biosensor) AND (infrared thermography OR IRT heat detection OR smart technology OR precision animal husbandry technology). The initial search yielded a total of 106 articles. The collected articles were thoroughly evaluated for their potential relevance by carefully reading the title, introduction, methodology, and discussion. In Figure 1, there is an example of a systematic literature review flow.

The following inclusion criteria were used to select eligible studies. First, only studies published as full-length, peer-reviewed scientific articles were considered eligible for inclusion. Second, the review was limited to studies related to estrus detection technologies for beef cattle but excluded studies conducted exclusively on dairy cattle. Third, relevant studies were required to assess the practical application of sensor-based monitoring systems, such as accelerometers, pedometers, boluses, and biosensors, and/or to investigate the use of infrared thermography (IRT) for reproductive management. Finally, only studies that provided appropriate technical performance indicators, such as sensitivity, and/or provided insights into the practical implementation of these technologies in cattle production systems were included in the review.

After applying the inclusion criteria and removing duplicate and irrelevant entries, 57 full-length articles were selected for this review. The selected studies were systematically analyzed and categorized based on several key aspects. Specifically, articles were classified according to the type of technology discussed (sensor-based systems and infrared thermography), the specific type of sensor used (e.g., accelerometers, pedometers, rumen bolus, biosensors, or thermal cameras), and reported performance metrics. In addition, each study was evaluated based on its reported practical advantages and challenges, and their current level of application in beef cattle management.

Notably, the review revealed that the number of studies specifically addressing the use of digital technologies for estrus detection in beef cattle is still limited. Compared to dairy cattle, this is a relatively new and emerging area of research. Further research is needed to validate and optimize the application of smart technologies in beef cattle farming systems, which often differ significantly in management practices and environmental conditions from intensive dairy operations.

## 3. Estrous Cycle Physiology

The estrous cycle encompasses the physiological and morphological changes occurring in cow reproductive systems from one estrus to the next. While these changes affect the entire organism, they are most pronounced in the reproductive organs. The estrous cycle is influenced by several factors, including nutrition, management practices, and housing conditions [8].

Under normal conditions, estrus recurs from every 17 to 24 days, although this interval may vary depending on a cow’s age and environmental conditions. A cycle is considered normal if estrus occurs no more frequently than every 17 days and no less than every 24 days. The regulation of the estrous cycle is primarily governed by the cerebral cortex and the pituitary gland, which secretes follicle-stimulating hormone (FSH) and luteinizing hormone (LH). FSH promotes follicular development and ovulation, initiates estrus, and enhances vascularization of the uterus. The growing dominant follicle produces estradiol, which triggers behavioral and physiological changes characteristic of estrus. Upon ovulation, the follicular rupture site forms the corpus luteum, which secretes progesterone, a hormone essential for pregnancy maintenance [9].

The behavioral manifestations of estrus in cows are defined differently by various researchers, depending on their methodological approaches. During estrus, cows exhibit increased restlessness, frequently sniffing and pressing their chin against other cows, displaying the flehmen response, and attempting to mount other animals [10]. They also demonstrate the characteristic “standing estrus” behavior, where they remain still while being mounted by other cows. The average duration of standing estrus in cattle ranges from 15 to 18 h, although the total estrus duration can vary significantly, spanning between 8 and 30 h. Cows in estrus typically exhibit mounting behavior between 20 and 55 times per cycle, with each mounting event lasting from approximately 3 to 7 s [11]. This behavior is predominantly observed during the nighttime period (6:00 p.m.–6:00 a.m.), during which up to 68% of all mounting events are recorded [12]. Additional behavioral changes include licking, rubbing, and increased aggression [13]. These behavioral signs are visually apparent and can be easily observed. However, certain estrus-related behavioral changes are more subtle and challenging to detect through visual observation, necessitating the use of advanced technologies. These smart technologies enable precise monitoring of increased activity levels in cows, including a rise in movement intensity and step count during estrus.

## 4. Sensor-Based Estrous Detection

In farm management, visual observation remains the most frequently used method for estrus detection. However, due to restricted cow movement and limited observation periods, a significant proportion of estrous events remain undetected. Research indicates that up to 40% of cows exhibit silent estrus within the first 60 days postpartum, where estrous signs are either absent or too subtle to be recognized, particularly in indoor housing systems. To overcome these limitations, automated estrus detection systems are increasingly being implemented, providing higher accuracy and reliability compared to traditional visual observation methods. These systems primarily rely on changes in physical activity, considered one of the most reliable estrus indicators. By integrating herd management and physiological monitoring technologies, optimized breeding and calving algorithms can be developed to suit specific cattle breeds. Moreover, the use of automated estrus detection allows for precise postpartum estrus monitoring, significantly reducing the interval between calving and insemination. Early conception at the beginning of the breeding season ensures that cows will calve at the start of the subsequent calving season, allowing for extended preparation time for the next reproductive cycle and resulting in heavier weaned calves. The adoption of automated estrus detection systems reduces labor input and time requirements compared to manual observation methods, facilitating the broader application of artificial insemination. This, in turn, accelerates genetic progress within herds, leading to a greater proportion of genetically superior and highly productive offspring. Recent advances in sensor technology and machine learning have significantly enhanced the accuracy of estrus detection by continuously collecting data on activity levels, rumination behavior and body temperature. Sensors and infrared devices generate valuable physiological and behavioral indicators associated with reproductive status. Algorithms can analyze these multi-parameter datasets to identify patterns of estrus. This integration of sensor-derived data reduces reliance on subjective visual observation and enables automated decision-making, and estrus can be detected earlier with greater precision, improving reproductive outcomes [14].

Currently, a variety of commercial solutions exist for monitoring livestock, each utilizing different sensor attachment methods. These devices cater to various applications, with some focusing solely on calving detection or digestive health tracking, while others, particularly those attached to the leg or collar, offer broader functionality. These multi-purpose sensors typically track movement and activity levels [15,16,17,18], with leg-mounted models also detecting the cow’s posture—whether standing, lying, or walking [19]. Moreover, estrus detection is a key feature of both collar and leg sensors, and certain models extend their capabilities to monitor rumination and feeding behavior.

Recent advancements in precision livestock farming highlight the power of artificial intelligence inclusion in farming and data analysis to improve monitoring programs in both dairy and beef cattle. Michelena et al. (2024) [20] reviewed how systems can integrate multiple sensor types, such as accelerometers, thermal cameras, and temperature boluses, to capture diverse behavioral and physiological signals associated with the estrus cycle. Processing the data gained through machine learning models and multimodal datasets enables robust estrus detection. Artificial-intelligence-driven sensor fusion enhances detection accuracy and supports real-time alerts for optimal insemination timing and improvements to reproductive outcomes in herd management [20].

### 4.1. Pedometer and Accelerometers

Pedometers have been shown to be particularly effective for estrus detection in beef cattle. For instance, Hojo et al. (2018) reported that a radiotelemetric pedometer could accurately detect estrus in Japanese Black cattle, indicating a notable increase in step count during both standing and silent heats [19]. This aligns with earlier findings by Yoshioka et al. (2010), who confirmed the effectiveness of pedometers in monitoring estrus through step count variations [21]. The ability of pedometers to identify heat periods accurately is crucial since timely insemination can significantly enhance reproductive success and efficiency in beef cattle farming [22].

Accelerometers, on the other hand, have allowed for a deeper understanding of cattle behavior and welfare. They also play a crucial role in estrus detection by capturing changes in behavior associated with the reproductive cycle. Richeson et al. (2018) discussed the significance of advanced technologies, including three axis accelerometers, in continuously monitoring cattle-related behaviors, which are pivotal in assessing health and detecting lameness or illness [22]. Another study by Robért et al. (2011) emphasized the utility of wireless accelerometers in determining lying behavior patterns, reduced resting time, and increased standing or walking, which are essential welfare indicators for healthy beef cattle and furthermore can serve as early indicators of estrus, especially over a significant period [23]. Similarly, Fromm et al. (2024) [24] detailed the application of ultrahigh-frequency (UHF) radio-frequency identification (RFID) systems that assess activity levels and lying behavior, demonstrating that consistent monitoring can lead to timely interventions for better management. It enables the identification of subtle deviations linked to estrus phases [24]. These findings support the integration of data from accelerometers into estrus detection protocols.

Numerous studies underscore the broader implications of using these technologies to enhance management practices. It has been established that advanced monitoring provides insights into feeding and watering behaviors, closely related to the overall health and productivity of beef cattle. This method of continuous assessment allows farmers to dynamically amend practices and address potential issues before they lead to significant losses [24,25,26,27,28].

Wearable sensors enable the real-time monitoring and analysis of cows’ physiological parameters on a large scale, making them essential tools for continuous reproductive status assessment. The application of these technologies provides an advantage over traditional estrus detection methods, as automated data collection and analysis eliminate subjectivity and ensure rapid and accurate decision-making that minimizes human error and subjectivity. Furthermore, the data recorded by wearable sensors are available in real time, allowing for immediate responses to detected changes, thereby optimizing estrus detection and insemination processes [29]. However, the effectiveness of these technologies largely depends on the type of sensor used. Different sensors monitor different parameters—accelerometers worn on the leg or neck are particularly effective in identifying increased physical activity and restlessness, both of which are common behavior signs of estrus. Neethirajan et al. (2020) [30] published how integrating diverse sensor types and combining them with wide-ranging data analytics and machine learning can fundamentally improve estrus detection in cattle. Diverse systems can continuously collect a range of behavioral and physiological data, analyze them, and identify the subtle yet consistent patterns associated with estrus onset. The research result is enhanced reproductive efficiency and improved breeding outcomes [30]. For example, the application of Smaxtec intraruminal boluses has been shown to effectively reduce the insemination index, shorten the calving-to-insemination interval, and improve reproductive efficiency. The focus has been on internal temperature and rumen activity, offering valuable insights into physiological changes that precede or coincide with estrus [28]. Additionally, the use of this technology has contributed to a decrease in abortion rates and a lower incidence of culling due to gynecological disorders, thereby enhancing overall herd fertility and reproductive performance [31]. Therefore, selecting the appropriate sensor type is crucial. In extensive beef production systems, where visual monitoring is limited, sensors must be robust, low-maintenance, and capable of functioning under variable environmental conditions. Strategic sensor deployment can significantly enhance reproductive outcome and improve decision-making in breeding programs [32]. Other examples of sensors used for estrus detection in beef cattle are presented in Table 1. The workflow of a sensor-based estrus detection system in cattle is presented in Figure 2.

### 4.2. Limination of Sensor Use and Animal Welfare Challenges

While precision livestock farming technologies offer clear benefits for estrus detection, they also present notable challenges that must be addressed. Sensor durability under harsh environmental conditions often faces issues, and scalable deployment in extensive beef cattle systems can be difficult to manage. Conventional precision farming systems depend heavily on cloud-based analytics, which can introduce significant latency and reduce reliability in remote environments. In regions with unstable power supplies and limited internet connectivity, sensor performance and data transmission are often compromised. Although solar-powered innovations offer potential solutions, their adoption may be limited by issues of accessibility or economic constraints.

Sensor placement in cattle itself presents several challenges, primarily due to the animals’ natural behavior and the necessity for regular human intervention for installation, adjustment, and maintenance. Cattle may dislodge or damage externally attached devices, making it difficult to ensure uninterrupted and reliable data collection [36]. Additionally, sensors are constrained by limited battery life, which restricts their effectiveness for long-term monitoring. Replacing or recharging these devices in large herds or remote locations can be both impractical and costly.

Furthermore, technical malfunctions are not uncommon, and the physical burden of the sensor hardware may affect animal comfort or behavior. Inaccurate estrus detection may occur due to insufficient field validation or the failure to incorporate meaningful behavioral or physiological indicators. There is also a risk of overdependence on technology, which may reduce farmer–animal interaction and diminish practical husbandry skills. In some cases, farm environments may be modified to suit sensor performance rather than to prioritize animal welfare. Addressing these limitations is essential to ensure that technological advancements genuinely support the health, productivity, and well-being of beef cattle [36,37,38].

## 5. Infrared Thermography

Infrared thermography (IRT) has emerged as a non-invasive, reliable, and efficient technology for monitoring physiological and reproductive parameters in beef cattle.

Early investigations by Wrenn et al. (1958) [34] examined thermal variations during the estrous cycle, laying the groundwork for further research in this area. Body temperature has commonly been monitored using rectally inserted digital thermometers, a technique that, while precise, may still induce animal discomfort and is affected by several procedural variables [39,40].

To overcome the limitations of traditional techniques, the use of advanced technologies such as infrared thermography has gained momentum. IRT utilizes thermal imaging to detect subtle temperature changes on the animal’s body surface, enabling the precise and stress-free monitoring of reproductive status. One of the most promising applications of infrared thermography (IRT) in cattle reproduction is estrus detection. During this phase, the autonomic sympathetic nervous system stimulates the release of catecholamines, which modulate vascular dynamics by inducing vasoconstriction and increasing blood flow, ultimately leading to changes in body temperature [41,42]. Enhanced blood circulation to the vulvar and perineal regions during estrus results in a localized temperature rise of approximately 1.3 °C, which can be effectively detected using IRT [43]. This alteration in temperature can be effectively measured using thermal cameras, thereby enabling the timely identification of cows in estrus. Studies show that the temperature in the vulvar area of cows in heat increases significantly more than that of cows that have not been in heat [44]. Such precise measurements significantly enhance the accuracy of artificial insemination timing, thereby increasing conception rates and reducing reproductive inefficiencies. Research indicates that body temperature tends to decrease roughly two days prior to the onset of estrus, followed by a sharp increase coinciding with the luteinizing hormone (LH) surge [45]. This sudden rise in temperature corresponds to an elevation in activity levels and the hormonal fluctuations typical of this phase of the estrous cycle. By effectively monitoring these thermal variations, predictive models can be established to ascertain ovulation timing, leading to superior insemination success rates. While various studies have tested methodologies for estrus detection using body temperature changes, it is essential to address the limitations posed by potential false positives linked to thermal measurements [46]. An integration of accurate breeding records, whether maintained manually or via automated software, can significantly fortify the reliability of estrus detection processes and amplify overall reproductive efficiency.

Beyond estrus detection, IRT also offers significant advantages in early pregnancy diagnosis. This technological advancement provides a non-invasive alternative to established methods such as transrectal ultrasonography. Thermal imaging conducted on the eyes, muzzle, flanks, and vulva regions showcases discernible temperature differences between pregnant and nonpregnant cows as early as 24 days post insemination [47]. Such capabilities enable improved reproductive decision-making, empowering farmers and veterinarians to strategize timely rebreeding protocols and enhance overall reproductive efficiency [48]. The utility of IRT transcends estrus identification and pregnancy evaluation, extending into the realm of diagnosing reproductive disorders. Conditions proliferating in beef cattle, such as metritis, endometritis, and ovarian cysts, can be detected by establishing elevated temperature patterns indicative of subclinical infections. The utilization of IRT in this context offers the potential for timely veterinary intervention, thus mitigating the impact of reproductive disorders on herd productivity [49]. Furthermore, IRT is being investigated for its capacity to assess sperm quality in bulls, where scrotal thermography may provide vital insights into testicular function and overall fertility potential [50].

While the benefits of IRT are substantial, certain limitations must be addressed to fully realize its potential in reproductive management. Environmental factors, such as ambient temperature and humidity, can influence thermal readings, necessitating the development of standardized protocols to mitigate these effects [51]. The integration of infrared thermography into beef cattle reproduction represents a considerable advancement in precision breeding technologies. By facilitating the early detection of reproductive events, aiding informed decision-making, and fostering improved herd productivity, IRT underscores a shift towards data-driven livestock management practices. As advances in smart technologies continue to progress, the future of beef cattle reproduction promises exciting possibilities with the ongoing application of IRT, AI-based analytics, and automated breeding management systems.

Furthermore, infrared thermography (IRT) demonstrates significant potential for estrus prediction, offering a means to reduce calving intervals and enhancing pregnancy rates following artificial insemination [52]. Comprehensive investigations across beef cattle have correlated physiological events associated with estrus with thermographically measured variations in body temperature, as summarized in Table 2. These studies collectively indicate that IRT can serve as a valuable tool, either independently or in conjunction with complementary methods, for estrus detection in beef cattle herds.

The integration of infrared thermography (IRT) is a significant advance in precision animal husbandry, providing greater accuracy in heat detection compared to traditional visual monitoring methods. However, its widespread adoption remains limited due to economic constraints, the requirement for specialized technical knowledge, and the influence of environmental factors on sensor reliability. Moreover, surface temperature measurements are significantly influenced by physiological factors such as blood perfusion, tissue metabolic rate, skin thickness, coat color, and fat layer insulation [56]. Environmental conditions—wind, solar radiation, temperature, debris—can also alter readings. To improve the practical application of IRT in beef production, future research should prioritize the improvement of sensor accuracy, integration of multi-sensor detection technologies, and development of cost-effective reproductive monitoring solutions. Operator expertise also plays a crucial role. These advances will facilitate the wider implementation of IRT, ultimately improving reproductive efficiency and herd management [57,58].

## 6. Conclusions

Accurate and rapid estrus detection remains a critical challenge in large-scale cattle production. Over the past decade, significant advancements in physiological monitoring technologies have transformed traditional livestock management. These automated technologies not only reduce human error in estrus detection but also provide real-time, objective physiological data, improving decision-making in reproductive management.

Accelerometers, pedometers, and biosensors are now widely used in beef cattle reproduction management. Research indicates that these technologies achieve estrus detection accuracies of 85–95%, depending on environmental conditions and sensor sensitivity, while IRT offers a contactless method by identifying temperature changes linked to the estrous cycle. Both approaches have shown the potential to improve artificial insemination success and reduce calving intervals.

Despite these advantages, the widespread implementation of precision estrus detection technologies faces economic and technical challenges. High equipment costs, data-processing complexity, and integration with existing farm management systems remain key barriers to adoption.

Further research is needed to develop economically affordable, user-friendly, and scalable solutions to enhance reproductive efficiency in beef cattle farming.

## Figures and Tables

**Figure 1 animals-15-02313-f001:**
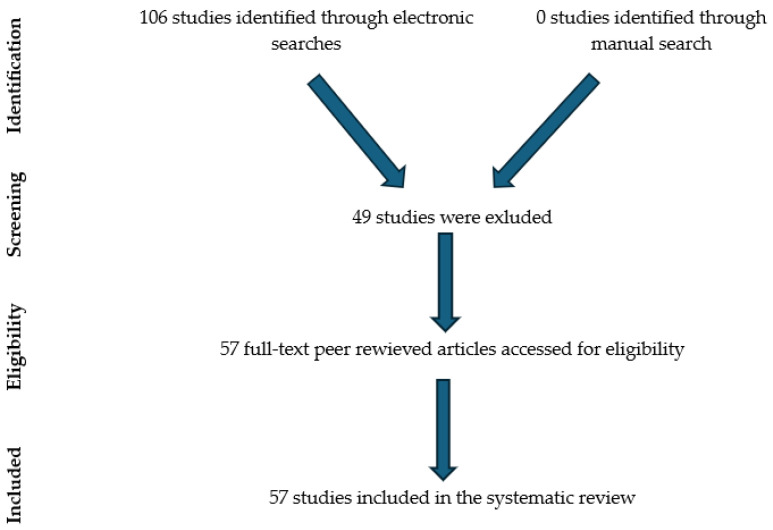
Systematic literature review flow diagram for study selection.

**Figure 2 animals-15-02313-f002:**
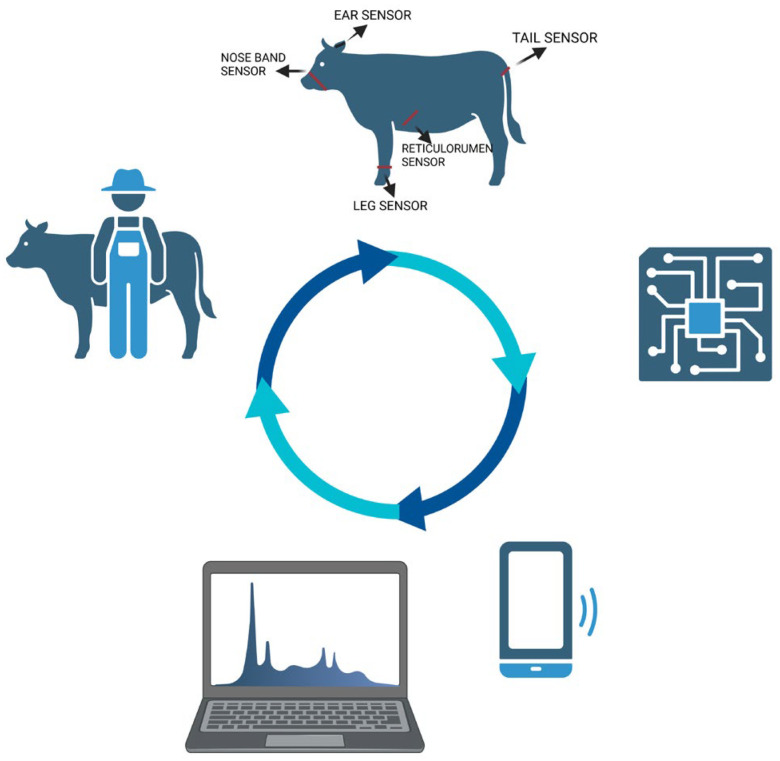
The workflow of sensor-based estrus detection system in cattle.

**Table 1 animals-15-02313-t001:** Sensor-based estrous detection systems.

Type of Sensor	Product Name, Company, Country	Sensitivity	Application	Parameters Measured	References
Accelerometer, Ear tag	HeatTime Pro+ (Allflex, Madison, WI, USA)	85–95%	Activity-based estrus detection	Physical activity, rest patterns	[16]
Accelerometer, Ear tag	CowManager (Agis, Harmelen, The Netherlands)	85–95%	Activity-based estrus detection	Physical activity, rumination	[33]
Accelerometer, Neck collar	Nedap (Nedap Livestock Management, Groenlo, The Netherlands)	79–94%	Estrus and health tracking	Physical activity	[34]
Accelerometer, Leg sensor	Gyuho (Comtech, Tokyo, Japan)	95%	Activity and estrus detection	Physical activity	[31]
Accelerometer, Neck collar	Heatime (SCR Engineers Ltd., Netanya, Israel)	85–95%	Estrus and reproductive health	Activity, rest patterns, feeding	[21]
Ruminoreticular biocapsule sensor	LiveCare (uLikeKorea, Seoul, Republic of Korea)	98–100%	Estrus and health tracking	Activity, body temperature	[18]
Multi-parameter Sensor, Neck collar	SenseHub Beef (Allflex Livestock Intelligence, Madison, WI, USA)	92%	Estrus detection, health monitoring	Activity, temperature, health	[33]
Thermometer, Ruminoreticular bolus	SmartStock (Smartstock, Boston, MA, USA)	-	Estrus and calving detection	Temperature	[17]
Biosensor, Ear tag	Moocall HEAT (Moocall Ltd., Limerick, Ireland)	88%	Estrus detection	Activity, health	[35]

**Table 2 animals-15-02313-t002:** IRT camera models used in various studies for estrus detection in cattle.

Reference	IRT Camera	Breed of Cattle	Site of IRT Observation	Conclusion
Radigonda et al., 2017 [51]	FLIR T300	Braford	Vulva	Infrared thermography was shown to be a reliable, non-invasive method for identifying estrus in Braford cows by detecting changes in vulvar temperature linked to ovarian activity. Cows in estrus displayed distinct temperature patterns compared to non-estrus cows. However, the accuracy of this technique can be affected by environmental conditions.
Ozaki et al., 2024 [52]	Video-based infrared camera ARGO P1-400	Japanese Black cows	Vulva, eyes, and pelvic area	The study demonstrates that monitoring ocular temperature is an effective, non-invasive way to predict ovulation in cows, even when typical signs of estrus are minimal. However, its reliability decreases in the presence of follicular cysts.
De Ruediger et al., 2018 [53]	FLIR E40	Murrah	Vulva, muzzle, and orbital area	The findings indicate that vulvar surface temperature is a consistent marker of hormonal changes linked to the estrous cycle in buffalo. In contrast, temperatures measured at the muzzle and around the eyes are more closely influenced by core body temperature and external environmental conditions.
George et al., 2014 [54]	FLIR ThermaCAM P65HS	Senepol	Eyes and muzzle	The study shows that eye temperature measurement using thermography offers a practical, non-invasive way to monitor cattle body temperature. Although cost currently limits widespread use, technological advancements and reduced prices could make it more accessible for routine health and welfare monitoring.
Kang et al., 2019 [55]	FLIR A615	Hanwoo	Topographic body surface	The study demonstrates that a thermal imaging-based tracking system can accurately identify estrus in Hanwoo beef cattle by monitoring behavioral changes such as increased activity and decreased feed intake. This approach offers a precise, non-invasive solution for enhancing estrus detection in beef herds.

## Data Availability

No new data were created or analyzed in this study.

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
