# Peer review of "The Role of Sensor Technologies in Estrus Detection in Beef Cattle: A Review of Current Applications"

_animals, 2025, doi:10.3390/ani15152313_

Round 1

Reviewer 1 Report

Comments and Suggestions for Authors

This is a very well written, comprehensive, and timely review of the subject matter. I do not have any editorial suggestions.

Author Response

Thank you very much for your kind and encouraging feedback. I'm glad to hear that you found the review comprehensive and timely. Your positive remarks are truly appreciated and motivating.

Reviewer 2 Report

Comments and Suggestions for Authors

In my opinion, this review is off-topic. The part associated with the review’s title is relatively short.

The first two sections describing estrus in beef cattle could be removed.

The authors could also add a figure showing the process of article selection.

Table 2, last column: please summarize the conclusions and not cite them directly.

In my opinion, the title should be changed: e.g. the role of sensors in heat detection in beef cattle.

l. 238 - 251 and others: author names are provided without references, e.g. Robert et al.

l. 241-246 This paragraph discusses the analysis of behavioral patterns of lying cattle, but how does this relate to oestrus? (this should be made clearer).

l. 253-259 – an off-topic paragraph generally discussing sensors.

l. 324 - 332 - an off-topic paragraph.

Perhaps the authors should cite the references by Suresh Neethirajan (“The role of sensors, big data and machine learning in modern animal farming”, Sensing and Bio-Sensing Research Volume 29, August 2020, 100367; https://doi.org/10.1016/j.sbsr.2020.100367), Tuyttens, F. A., Molento, C. F., & Benaissa, S. [(2022). Twelve threats of precision livestock farming (PLF) for animal welfare. Frontiers in Veterinary Science, 9, 889623], Benaissa, S., Tuyttens, F. A. M., Plets, D., Trogh, J., Martens, L., Vandaele, L., ... & Sonck, B. [(2020). Calving and estrus detection in dairy cattle using a combination of indoor localization and accelerometer sensors. Computers and electronics in agriculture, 168, 105153.], Álvaro Michelena, Óscar Fontenla-Romero, José Luis Calvo-Rolle [, A review and future trends of precision livestock over dairy and beef cow cattle with artificial intelligence, Logic Journal of the IGPL, 2024;, jzae111, https://doi.org/10.1093/jigpal/jzae111] and Shine, Philip, and Michael D. Murphy. ["Over 20 years of machine learning applications on dairy farms: A comprehensive mapping study." Sensors 22.1 (2021): 52.].

It would also be useful to enrich the manuscript with drawings, diagrams or figures in general that would explain the method of sensor placement, the flow of information etc.

The authors should also discuss the disadvantages of using sensors in more detail. Also, the errors generated by such sensors should be described. In general, some discussion of sensor limitations should be provided.

Also, the use of thermal imaging has some important limitations. These limitations should also be discussed in more detail, e.g. the surface temperature values obtained by IRT depend on the quantitative influence of the environmental conditions and the thermoregulatory response of the animal. In addition to the amount of blood perfusion, skin temperature depends on the intensity of tissue metabolism, the type and color of the hair coat and the thickness of adipose tissue. The reliability of a thermal imaging study depends on the technical parameters of the cameras, environmental conditions, the experience of the operator, the individual characteristics of the animals and the methodology of the study. Since many factors can affect the surface distribution of an animal's body temperature, and thus the result of a thermographic measurement, the influence of any stimuli that interfere with the measurements should be minimized during thermal imaging studies. In addition, in order to reduce the risk of image misinterpretation, it would be necessary to use so-called protocols for normalizing imaging parameters, i.e. standards that would allow to receive reliable test results. The main limitation in implementing these standards in livestock thermography is the inability to compare thermograms taken under different environmental conditions.

Author Response

  1. “This review is off-topic. The part associated with the review’s title is relatively short.”
    Response: Thank you for this observation. We have restructured the manuscript to better align with the original focus. Specifically, we have expanded the sections directly addressing the role of sensor technologies in estrus detection in beef cattle and removed or condensed content that was more general in nature. Additionally, we have revised the title (see below) to reflect the main scope more accurately.

  1. “The first two sections describing estrus in beef cattle could be removed.”
    Response: Thank you for this observation. These sections have been removed.

  1. “The authors could also add a figure showing the process of article selection.”
    Response: Thank you for this observation. A PRISMA-style flow diagram has been added to illustrate the process of article identification, screening, and inclusion in the review. Systematic literature review flow diagram is shown in Chart 1.

  1. “Table 2, last column: please summarize the conclusions and not cite them directly.”
    Response: Thank you for pointing this out. We have revised the final column of Table 2 to provide concise summaries of the studies' conclusions in our own words.

  1. “The title should be changed.”
    Response: Thank you. We agree and have revised the title to: The Role of Sensor Technologies in Estrus Detection in Beef Cattle: A Review of Current Applications

  1. “l. 238–251 and others: author names are provided without references.”
    Response: Thank you for pointing this out. All authors mention throughout the manuscript have now been accompanied by appropriate in-text citations and added to the reference list to ensure consistency and traceability.

  1. “l. 241–246: clarify how lying behavior relates to estrus.”
    Response: Thank you for pointing this out. This paragraph was adapted and revised 247-275.

  1. “l. 253–259 – an off-topic paragraph discussing sensors.”
    Response: This paragraph has been removed to maintain focus on estrus detection and to avoid broad generalizations not directly tied to the manuscript’s scope.

  1. “l. 324–332 – an off-topic paragraph.”
    Response: This paragraph has also been removed for the same reason above.

  1. “Consider citing the following references...”
    Response: Thank you for the excellent suggestions. We have reviewed all the recommended references and incorporated them where appropriate to enrich the discussion, especially in relation to sensor limitations, welfare implications, and machine learning applications.

  1. “Add drawings/figures explaining sensor placement, information flow, etc.”
    Response: Thank you for the excellent suggestions. We have added a schematic diagram illustrating common sensor placement strategies in cattle, Picture 1.

  1. “Discuss disadvantages of using sensors, including errors and limitations.”
    Response: Thank you for pointing this out. A dedicated subsection has been added to address the drawbacks and limitations of sensor use, 284-300, 426-434.

  1. “Discuss the limitations of thermal imaging in more detail.”
    Response: Thank you for the excellent suggestions. We fully agree and have significantly expanded the section on thermal imaging 391-403.

We are grateful for your insightful feedback, which has greatly improved the focus, clarity, and scientific value of our review.

Reviewer 3 Report

Comments and Suggestions for Authors

Dear Authors,

The manuscript addresses a highly relevant topic - the use of modern digital technologies, in particular sensor-based systems and infrared thermography (IRT), to improve reproductive efficiency in beef cattle farming. The solutions highlighted in the paper are indeed promising for enhancing estrus detection practices, and the authors demonstrate a solid understanding of the physiological background and technical potential of these technologies. The material is presented in a clear, accessible, and well-structured manner, which contributes positively to the overall impression.

However, to qualify as a review article, the manuscript requires substantial revision in the following areas:

- The article is based on 61 references, which is insufficient for a systematic review. A significant proportion of the cited sources focus on dairy cattle or general animal science topics, whereas the declared focus - beef cattle production - is addressed selectively. The bibliography should be expanded to include more specialized sources that analyze the specific challenges and management practices in beef cattle.

- The specifics of beef cattle production systems are largely overlooked. In most cases, these systems are extensive (e.g., pasture-based), with minimal infrastructure. The implementation of automated technologies in such conditions faces substantial barriers (e.g., power supply, internet connectivity, cost, maintenance), which are not adequately addressed. This is a critical gap, particularly given the applied nature of the topic.

- The article does not analyze the limitations of sensor use under real-world conditions in beef herds. There is no clear evidence on how thoroughly these technologies have been validated in beef cattle, which are typically managed with less human interaction and greater environmental exposure than dairy cows.

- The manuscript presents the material in a predominantly linear and descriptive fashion, listing available technologies without critical synthesis or comparative conclusions. A stronger analytical structure is recommended, with comparative evaluation of different approaches (e.g., pedometers vs. IRT) and discussion of their strengths and limitations under beef cattle conditions.

In summary, the topic is worthy of attention and the article has a strong foundation. However, in its current form, the manuscript falls short of expectations for a comprehensive and critical review. After expanding the literature base, incorporating the specifics of the beef sector, and refining the analytical framework, the article has the potential to make a valuable contribution to the field of precision livestock farming.

With respect,

Comments on the Quality of English Language

The manuscript is written in clear and accessible language. However, a minor editorial revision by a native English speaker would further enhance its readability.

Author Response

We would like to sincerely thank the reviewer for the thoughtful and thorough evaluation of our manuscript. We also value the detailed suggestions for improvement, which have guided us in significantly revising and enhancing the manuscript. Below, we address each point individually. We adapted and revised our all review, also we have changed the title.

  1. “The article is based on 61 references, which is insufficient for a systematic review...”
    Response: We fully acknowledge this limitation. In the revised manuscript, we have substantially expanded the reference list to include a broader and more representative range of peer-reviewed articles specifically addressing estrus detection and reproductive management in beef cattle and removed not related, off-topic references.

  1. “The specifics of beef cattle production systems are largely overlooked...”
    Response: Thank you for highlighting this crucial point. We have added a new sections and expanded almost all paragraphs in order to expand and analyze sensors more deeply.

  1. “The article does not analyze the limitations of sensor use under real-world conditions in beef herds...”
    Response: Thank you for this observation. We added few paragraphs and analyzes the limitations of sensors 284-300, 426-435.

  1. “The manuscript presents the material in a predominantly linear and descriptive fashion...”
    Response: We appreciate this valuable insight. The revised manuscript now follows a more analytical and comparative structure.

We believe these changes have significantly strengthened the manuscript and brought it in line with the expectations for a critical and comprehensive review.

With respect and appreciation,
The Authors

Round 2

Reviewer 2 Report

Comments and Suggestions for Authors

The numbering of manuscript sections is incorrect. Please revise.

l. 98: ICT or IRT?

Please define the abbreviations at their first mention in the text and use them consistently throughout the manuscript.

l. 100: The initial number of papers (101) does not correspond to the number in Figure 1 (106). Please clarify.

l. 109: : ITC or IRT?

l. 128 and 280: Please use “Figure” instead of “Chart” and “Picture” and number the figures consecutively in the whole manuscript.

l. 239 – 246: The off-topic paragraph is still present in the revised version of the manuscript.

l. 181 and 365: Please do not use the same abbreviation (AI) both for artificial insemination and artificial intelligence. It is misleading.

l. 373-380: This paragraph is an exact repetition of lines 359-366. Please remove it.

Table 2: Please use “et al.” instead of “et Al.”.

l. 399 and 402: ICT or IRT?

l. 429 -434: Please move this paragraph to Section 5.2.

Conclusions: Please shorten this section. Currently, it is too long. Please summarize only the most significant findings resulting from your study.

References: Please format the references according to the journal requirements.

Comments on the Quality of English Language

The quality of English language must be improved (especially in the paragraphs included in the revised version).

Author Response

  1. “The numbering of manuscript sections is incorrect. Please revise.“
    Response: Thank you for pointing this out. We have carefully revised the manuscript and corrected the numbering of all sections to ensure logical consistency and proper formatting throughout the document.
  2. “l. 98: ICT or IRT? Please define the abbreviations at their first mention in the text and use them consistently throughout the manuscript.“
    Response: Thank you for your comment. We have reviewed line 98 and clarified that the correct abbreviation is IRT (Infrared Thermography). The abbreviation has now been defined at its first mention in the text and used consistently throughout the manuscript. We appreciate your attention to detail.
  3. “l. 100: The initial number of papers (101) does not correspond to the number in Figure 1 (106). Please clarify.“
    Response: Thank you for noticing this discrepancy. We have reviewed the literature selection process and corrected the inconsistency. The initial number of articles has been updated to 106 in the text to match Figure 1. This change ensures consistency and accuracy in the description of the review methodology.
  4. “l. 109: : ITC or IRT?“
    Response: Thank you for pointing this out. The correct abbreviation is IRT (Infrared Thermography). We have corrected the term at line 109 and ensured consistent use of the abbreviation throughout the manuscript.
  5. “l. 128 and 280: Please use “Figure” instead of “Chart” and “Picture” and number the figures consecutively in the whole manuscript.“
    Response: Thank you for your suggestion. We have replaced the terms “Chart” and “Picture” with “Figure” at lines 128 and 280. Additionally, all figures have been numbered consecutively throughout the manuscript to ensure consistency with journal formatting guidelines.
  6. “l. 239 – 246: The off-topic paragraph is still present in the revised version of the manuscript.“
    Response: Thank you for your comment. We agree with your concern and confirm that the off-topic paragraph in lines 239–246 has been removed in the revised version to maintain the focus and relevance of the manuscript.
  7. “l. 181 and 365: Please do not use the same abbreviation (AI) both for artificial insemination and artificial intelligence. It is misleading.“
    Response: Thank you for pointing this out. We fully agree that using the same abbreviation (AI) for both artificial insemination and artificial intelligence may cause confusion. We have revised the manuscript to eliminate ambiguity by using the full term artificial insemination and abbreviating only artificial intelligence as AI. This distinction has been applied consistently throughout the text.
  8. “l. 373-380: This paragraph is an exact repetition of lines 359-366. Please remove it.“
    Response: Thank you for bringing this to our attention. We acknowledge the repetition and have removed the duplicated paragraph at lines 373–380.
  9. “Table 2: Please use “et al.” instead of “et Al.”.“
    Response: Thank you for your observation. We have corrected the formatting in Table 2 and now use the proper lowercase form “et al.” consistently throughout the table.
  10. “l. 399 and 402: ICT or IRT?“
    Response: Thank you for pointing this out. We have reviewed the text and confirmed that the correct term is IRT (Infrared Thermography). The abbreviation has been corrected at lines 399 and 402, and consistent usage has been ensured throughout the manuscript.
  11. “l. 429 -434: Please move this paragraph to Section 5.2.“
    Response: Thank you for your suggestion. We have moved the paragraph from lines 429–434 to Section 4.2 (formerly 5.2), as requested.
  12. “Conclusions: Please shorten this section. Currently, it is too long. Please summarize only the most significant findings resulting from your study.“
    Response: Thank you for your suggestion. We have carefully revised and shortened the Conclusions section by removing repetitive information and summarizing only the most significant findings. The updated version highlights key technologies, their effectiveness, and the main barriers to implementation, while maintaining clarity and scientific relevance.
  13. “References: Please format the references according to the journal requirements.“
    Response: Thank you for your feedback. We have revised the reference list to align with the journal’s formatting requirements. Please let us know if any additional adjustments are needed.
  14. “The quality of English language must be improved (especially in the paragraphs included in the revised version)."
    Response: We would like to clarify that the manuscript has undergone professional English language editing through MDPI Author Services, as confirmed in the attached communication. The revised version reflects these improvements.

We are grateful for your insightful feedback, which has greatly improved the focus, clarity, and scientific value of our review.

Reviewer 3 Report

Comments and Suggestions for Authors

Dear Authors,

Thank you for your efforts in improving the manuscript. I appreciate the detailed responses and substantial revisions, including the extended reference list, refined structure, and added analysis of sensor limitations. The manuscript now offers more depth and critical insight. I support its publication.

Sincerely,

Comments on the Quality of English Language

Overall, the manuscript is written in clear and accessible English. Nevertheless, a careful language revision by a native speaker or professional editor is recommended to further enhance clarity and consistency throughout the text.

Author Response

Dear Reviewer,

Thank you very much for your positive feedback and support for the publication of our manuscript. We truly appreciate your constructive comments and recognition of our efforts to improve the content, structure, and depth of the work.

Regarding the English language, we would like to confirm that the manuscript has been professionally edited by MDPI Author Services. We hope that this ensures the clarity and consistency expected. 
